# Exploring the use of art interventions in challenging stigmas related to neurological disorders: A scoping review

Jack Lumsdon[1]☉, Vivian Nyadimo[2]☉, Shilla Dama Unda[2], Natasha Fothergill-Misbah[1‡], Mary A. Bitta[2,3,4,5]*‡

1 Population Health Sciences Institute, Faculty of Medical Sciences, Newcastle University, Newcastle Upon Tyne, United Kingdom, 2 Brain and Mind Institute, Aga Khan University, Nairobi, Kenya, 3 Department of Global Health and Social Medicine, Harvard University, Boston, United States of America, 4 Department of Psychiatry, University of Oxford, Oxford, United Kingdom 5 Faculty of Liberal Arts and Professional Studies, York University, Totonto, Canada

☉ These authors contributed equally to this work.
‡ Joint last authors to this work.
* tash.fothergill-misbah@newcastle.ac.uk

## Abstract

### Introduction

Stigma remains a major barrier to accessing healthcare for people with neurological disorders. The World Health Organization's Intersectoral Global Action Plan on epilepsy and other neurological disorders urges all member countries to have at least one functioning awareness campaign for neurological disorders by 2031. Art has emerged as a valuable tool for social change with previous success in highly stigmatised disorders. This review aims to understand the effectiveness of art-based interventions in reducing stigma related to neurological disorders.

### Methods

A scoping review was conducted on three databases (PubMed, Web of Science, and Embase); from inception to October 2024. Data relevant to the study's aims were extracted and underwent narrative synthesis to develop key themes and patterns across the included studies.

### Results

After removing duplicates, 9,992 articles were screened with 24 articles identified for inclusion. Studies were predominantly from high-income settings (63%) and addressed stigma in dementia (38%), epilepsy (42%), and stroke (21%). Included studies targeted young children through to older adults. The most common form of art was videos (38%) which were predominantly short and educational, followed by visual arts (42%) which included short films, adverts, and use of images, and dance

**Data availability statement:** All relevant data are within the paper and its Supporting information files.

**Funding:** This project was funded by the British Academy grant number OIIRP230192.

**Competing interests:** The authors have declared that no competing interests exist.

and theatre (25%). Art was used to change attitudes and perceptions (71%), raise awareness and enhance knowledge (54%), and influence emotions and behaviour (46%).

## Conclusions

This review highlighted that art-based interventions effectively contributed to stigma reduction by addressing misinformation, challenging prejudices, and encouraging supportive actions but data from low resource settings were scarce. Interventions prioritising the direct interactions between a small number of the public and people with lived experience seemed particularly effective, however, the potential for theatre productions and digital media to target a larger audience should be considered.

## Introduction

Neurological disorders affect over 1 in 3 people and are the leading cause of illness and disability worldwide [1]. This burden is felt the most in low- and middle-income countries (LMICs), where 70% of people with neurological disorders live [2]. Despite advancements in the diagnosis and treatment of neurological disorders, multiple barriers limit the accessibility of health services. One significant barrier to the diagnosis, treatment, and social integration of individuals with neurological disorders is stigma [3]. Stigma associated with neurological disorders is a widely recognised global health issue [4]. Stigma can have many cumulative downstream consequences which can have direct health implications, for example, delayed diagnosis and limited access to treatment, as well as social implications, including social isolation, thereby exacerbating the disease burden and hindering effective care [5]. Understanding and mitigating stigma surrounding neurological disorders is essential for improving the wellbeing of individuals and enhancing quality of life.

Stigma related to neurological disorders takes multiple forms including social, self, and structural. Social stigma leads to exclusion and discrimination [6], self-stigma causes shame and deters care-seeking [5]., and structural stigma is embedded in policies, which can reduce access to treatment [7]. Stigma is often more pronounced in LMICs due to varying levels of awareness and medical education, which is largely a result of little funding, insufficient human resources and a lack of political support to invest in anti-stigma campaigns [8]. While some stigmatising factors such as fear, visibility, and unpredictability of symptoms may be globally experienced [9]. Some factors can be continent specific such as cultural beliefs around witchcraft and curses which are common in African countries such as Tanzania and Kenya for disorders like epilepsy and Parkinson's [10–12]. Other continents, such as Asia, see a diagnosis as private and would prefer to be labelled drunk rather than divulge a diagnosis [13]. Both of which negatively influence health seeking behaviours and contribute towards isolation. Sudden or fluctuating symptoms such as seizures or stroke may provoke fear, misattribution, and isolation [14,15]. This complex stigma landscape hampers inclusion, care, and understanding for individuals living with neurological disorders, perpetuating cycles of fear and exclusion [16].

Tackling stigma is a key goal of the World Health Organization's Intersectoral Global Action Plan on epilepsy and other neurological disorders 2022–2031 (IGAP) [2], which outlines a proposed action for Member States to *"lead and coordinate intersectoral advocacy strategies for reducing stigma and discrimination"* (p. 14). IGAP's global target 1.2 states that 100% of countries will have at least one functioning awareness campaign of advocacy programme for neurological disorders by 2031. Although various awareness campaigns for neurological disorders have launched globally, public education remains inadequate, and stigma continues to limit access to healthcare, employment, and social inclusion for affected individuals [11,17]. It is unclear what these awareness campaigns should look like, particularly in LMICs, and how to measure their success. Creating a functional campaign requires innovative strategies and active engagement of stakeholders across sectors, ensuring the empowerment and involvement of people with lived experience of neurological disorders including their caregivers and members of the communities they live in. Given these challenges, it is critical to explore alternative, culturally relevant approaches that can help shift perceptions, reduce stigma, and promote inclusion.

Art has emerged as a valuable tool for social change, offering a platform to challenge stereotypes, foster empathy, and promote dialogue [18,19]. Within healthcare, artistic interventions serve both educational and therapeutic functions, enabling individuals to share their lived experiences and humanise stigmatised disorders [20]. The use of arts has also been successful in tackling the stigma of certain disorders such as mental illness [21]. Various art based initiatives are now branching into neurological disorders to provide compelling representations of the experiences of people with lived experience which serve to enhance public understanding and engagement [22]. These initiatives have been useful in challenging societal stigma and structural stigma by advocating for a holistic approach to care which values meaningful engagement. Performing arts offer interactive and immersive storytelling approaches that can highlight the struggles and resilience of people with lived experience, thereby countering prevailing misconceptions [23]. Similarly, literary arts and visual arts, serve as effective mediums for conveying the realities of people with lived experience [24], such as the use of photo-stories and fiction writing to address HIV stigma [25].

The potential use of art in addressing stigma related to neurological disorders remains largely unexplored [26]. This review seeks to assess the effectiveness of art-based interventions in reducing stigma related to neurological disorders, identify the most impactful artistic approaches and their mechanisms of influence, examine how demographic and cultural factors shape intervention effectiveness, and evaluate the challenges, barriers, and long-term sustainability of art-based stigma reduction initiatives.

## Methods

This review aims to provide a general overview of the literature, as opposed to answering a specific question. As such, a scoping review was deemed the most appropriate methodology [27]. This scoping review was conducted in accordance with the Preferred Reporting Items for Systematic Reviews and Meta-Analyses extension for Scoping Reviews (PRISMA-ScR) guidelines [28]. A completed PRISMA checklist is provided as supplementary S1 File. The aim of the review is to systematically map existing literature on the use of art-based interventions for stigma reduction in neurological disorders, a protocol was not published for this.

### Eligibility criteria

This review of peer-reviewed original publications included studies that evaluated interventions incorporating visual arts theatre, film, storytelling, or music as a means of addressing stigma. Studies were eligible if they assessed the impact of these interventions on stigma-related outcomes (conceptualised as changes in knowledge, attitudes and behaviour) and provided information on intervention characteristics, target populations and geographic distribution. Published reviews were excluded, although reference lists were searched for any relevant articles. No restrictions were imposed on study design, publishing date or geographic location to ensure a comprehensive synthesis of the available evidence.

## Information sources

Systematic searches were conducted across three key databases: PubMed, Web of Science, and Embase. These databases were chosen as they provide a comprehensive and multi-disciplinary overview of available literature. PubMed highlights medical research while Web of Science and Embase provide access to social science literature. Searches yielded relevant peer-reviewed literature published from inception until October 2024 when the search was carried out.

## Search strategy

The search strategy identified articles using art-based interventions in the context of neurological disorders targeting stigma and efforts to raise awareness. The search strategy can be found in S2 File.

## Selection process

All search results were imported into Rayyan, a web-based application designed to assist in systematic reviews [29]. Duplicate records were identified using Rayyan then manually checked and removed. Two independent reviewers (VN & SD) conducted the title and abstract screening process for relevance, followed by a full-text review by VN & JL for eligible studies. Citation tracking was used to identify additional relevant articles which were not identified within the initial search. Discrepancies in study selection were resolved through discussion or consultation with a third reviewer (NFM & MB).

## Data extraction

A data extraction form was developed to include relevant data items to address the review's aims. To ensure rigor and reduce bias, data extraction was performed by two reviewers (VN & JL) to collect information on intervention type, study design, target population, geographic distribution, and stigma-related outcomes; any discrepancies were discussed and resolved through consensus.

## Data interpretation

Using the Patterns, Advances, Gaps, Evidence for practice and Research recommendations framework, the key findings of the review were visualised using a patterning chart [30]. Extracted data were analysed using a narrative synthesis approach to identify key themes and patterns across studies [31]. Given the expected heterogeneity in study methodologies and outcome measures, no formal meta-analysis was conducted. Instead, findings were summarised descriptively, with tabulated data where applicable, to provide a structured overview of intervention characteristics and effects.

## Results

### Study characteristics

As indicated in Fig 1, three databases were searched for articles published prior to October 2024. A total of 12,420 articles were identified. After removing duplicates, 9,992 titles and abstracts were screened. Fifty-two articles were retrieved for a full text review, with 24 articles identified for inclusion within this review. Extracted study characteristics (Table 1) and an overview of extracted data (Table 2) enabled patterns within the data to be summarised. The final extracted dataset is available as supplementary S3 File.

Included studies predominantly occurred in high-income countries in North America, Europe, and Australasia (63%) [34,35,39–48,51–53] with fewer studies conducted across the South Americas (13%) [36,50,54], Asia (17%) [32,33,38,49], and Africa (13%) [37,49,55]. Art interventions aimed at reducing stigma addressed three neurological disorders: dementia (38%) [41–45,48,51–53], epilepsy (42%) [32,33,36,37,39,40,46,47,54,55], and stroke (21%) [34,35,38,49,50]. Interestingly, no articles were included tackling other common neurological disorders such as Parkinson's disease and Multiple Sclerosis.

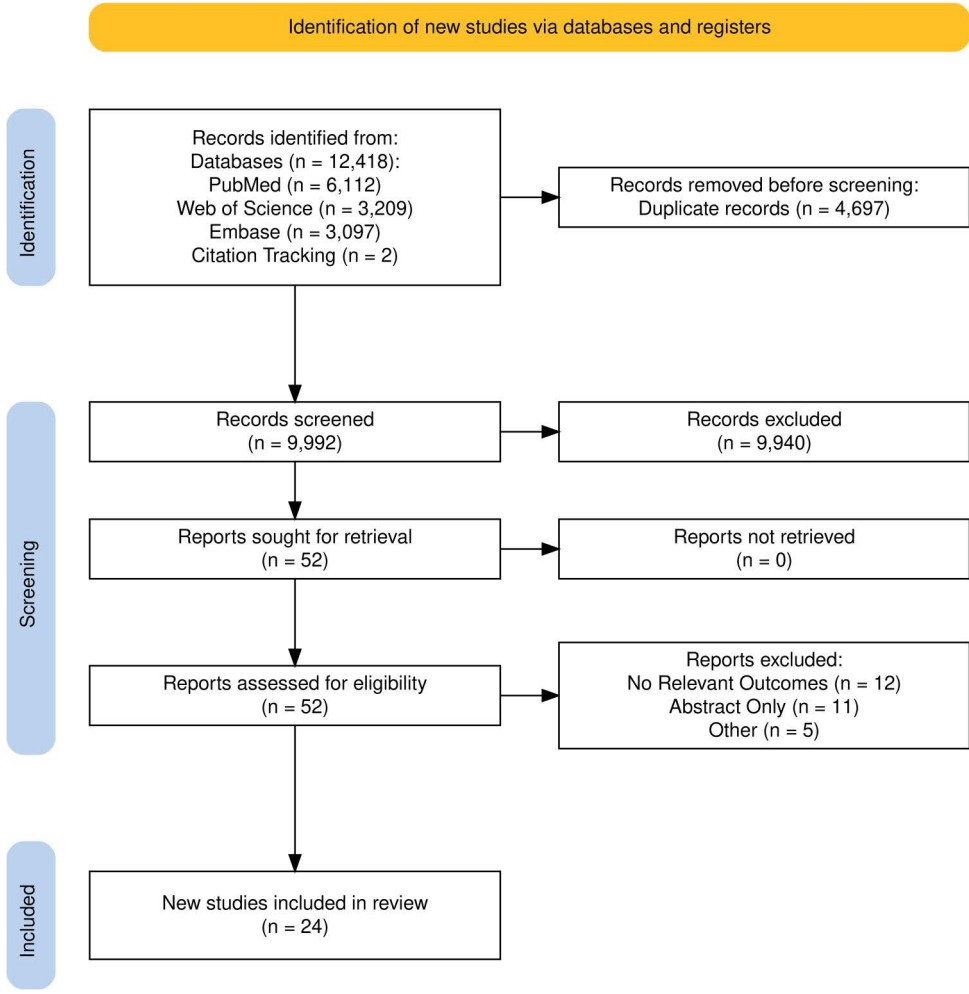

**Fig 1. PRISMA flow diagram of included studies.**

### Targeted populations and types of stigmas explored

The target population of the art intervention varied between the general population (50%) [32–34,36,38,41,47–52], children (33%) [32,35,37,39,40,53–55], and university students (25%) [42–47]. Additionally, some papers explored internal stigma among people with lived experience (13%) [36,41,52]. 71% articles reported the age of participants with a range of 7–65+ [32,34,35,37,39,40,43,45–51,53–55] while 67% reported the sex of participants, with the participation of females ranging from 27–85.3% with an average of 56% females per study [33,34,37,39–43,45–51,54,55]. Due to the broad methods of data collection, some studies did not report demographics.

While all studies explored the use of arts to address stigma, hence their inclusion, these could be broken down further into studies exploring the effectiveness of educational materials (46%) [32,33,35,37,39,40,46,49,50,54,55], focusing on leveraging the lived experience to reduce stigma (38%) [36,41–45,51–53], and a mix of the two aims (17%) [34,38,47,48].

### Types of art employed

The most common form of art employed was videos (38%) [32–35,40,41,46,49,50]. These were predominantly short educational videos conveying disorder specific information such as risks, symptoms, and how to respond to a stroke

**Table 1. Study characteristics of included studies.**

| Author | Aim | Country | Participants/sample size | Data collection method | Art used/ Medium of delivery/level of engagement/ campaign frequency | Analysis techniques | Key findings |
|---|---|---|---|---|---|---|---|
| Fong et al. (2018) [32] | Assess the baseline level of AKA among Malaysian school teachers and students and assess the effectiveness of the IAEEP in raising AKA of epilepsy among Malaysian teachers and students | Malaysia | 121 Sample Size (54 Teachers & 67 Secondary School Students aged 13–19) | Awareness, Knowledge, and Attitude (AKA) Epilepsy Questionnaire | Animation by video through active participation in a one off campaign | Statistical | Significant increase in awareness post-intervention (e.g., have you heard or read anything about epilepsy, have you seen anyone having an epilepsy attack). Significant increase in knowledge attitude post-intervention (e.g., do you think epilepsy is infectious, caused by evil spirits, do you think people with epilepsy can drive, have a family). |
| Fong et al. (2019) [33] | Assess the baseline level of AKA among Malaysian parents of children with epilepsy, explore potential characteristics associated with low levels of AKA, assess the effectiveness of the IAEEP in improving parental AKA and whether these changes remain stable over time, and explore the effects of the IAEEP on parental mental state and perception of their child's quality of life | Malaysia | 78 Sample Size (Parents of children with epilepsy, 66% female) | Awareness, Knowledge, and Attitude (AKA) Epilepsy Questionnaire & Health-Related Quality of Life Measurement for Children with Epilepsy (CHEQOL-25) Questionnaire & Depression, Anxiety, and Stress Scale (DASS-21) Questionnaire | Animation by video through active participants participation in a one off campaign | Statistical | Significant increase in awareness post-intervention (e.g., have you heard or read anything about epilepsy, have you seen anyone having an epilepsy attack). Significant increase in knowledge attitude post-intervention (e.g., do you think epilepsy is infectious, caused by evil spirits, do you think people with epilepsy can drive, have a family). |
| Haesebaert et al. (2020) [34] | Assess the impact of the campaign on the number of calls to emergency medical services for suspected stroke and to assess the effect of the campaign on time from symptom onset to emergency medical service call and the public's knowledge of stroke in the intervention county compared to a control county | France | 1,421 Sample Size (General Population) | Emergency Medical Service calls, Attitude and knowledge ad-hoc questionnaire | Visual Arts, Theatre, Leaflets, Radio and Television Plays, all co-produced with two patient representatives and distributed via digital platforms, in person events, online galleries and posters through passive participants participation in an on going campaign | Statistical | In the intervention county, proportion of the population with knowledge of at least 2 stroke symptoms and understanding that everyone can be affected by stroke significantly increased. Significant increase in emergency medical services calls in reaction to stroke symptoms in the intervention county compared to the control county. |
| Mikulik et al. (2019) [35] | Evaluate the feasibility and effectiveness of an innovative school stroke and heart attack educational programme for children | Czechia | 3,336 Sample Size (Children aged 13–15) | Disease knowledge and correct response questionnaire | Film by in person event through active participation in a one off campaign | Statistical | Significant increase in correct responses for stroke and heart attack which remained at 3–6 month post-test, significant increase in knowledge for stroke post-intervention but not for heart attack. |
| Spagnol et al. (2019) [36] | Demonstrate through arts the research findings on emotional burden related to epilepsy | Brazil | People with Epilepsy, caregivers, volunteers, and researchers | Comments from audience | Dance by performances through passive participation in a one off campaign | Thematic | Patients felt valued and represented when viewing the dance, they related to several scenes. |

*(Continued)*

**Table 1.** (Continued)

| Author | Aim | Country | Participants/sample size | Data collection method | Art used/ Medium of delivery/level of engagement/ campaign frequency | Analysis techniques | Key findings |
|---|---|---|---|---|---|---|---|
| Tekle-Haimanot et al. (2016) [37] | To assess the impact of the comic book on epilepsy-related knowledge, attitude, and practice among school age children in both rural and urban settings | Ethiopia | 226 Sample Size (High school students aged 16–20+, 52% female) | Knowledge, Attitudes and Practice (KAP) Questionnaire | Comics by in person event through passive participation in a one off campaign | Statistical | Significant improvement in knowledge regarding epilepsy, such as cause of epilepsy, whether it is contagious, treatment options, and how to respond to a seizure attack. Significant improvement when asked whether they would allow a sibling to marry someone with epilepsy |
| Zhao & Liu (2021) [38] | To wake up not only potential stroke patients, but also the societies to raise awareness for stroke and the elderly who live alone, thus being part of the community's efforts to prevent stroke-related disasters | China/ Global | General population | Movie reviews | Film by use of digital platforms through passive participation and remains freely available for all to view. | Thematic | Raises awareness of how to identify and respond to a stroke. One comment talks about how they could share the film with a parent to spread awareness. The movie was able to deliver practical information to educate and help people that would be understood by a lay audience. The movie was regarded as powerful and very impactful, it was very touching and brought tears to people. |
| Brabcová et al. (2017) [39] | To evaluate the effectiveness of two interventions focused on the reduction of epilepsy-related stigma in children aged 9–11 years | Czechia | 182 Sample Size (Children aged 9–12 with an average age of 10, 53% female) | Stigma Scale of Epilepsy (SSE), Knowledge of epilepsy questionnaire | Film or Story telling by in person event through passive participation in a one off campaign | Statistical | For both the educational video and educational story there was a significant improvement in knowledge of Epilepsy, with the story seemingly slightly better. For both the educational video and educational story there was a significant decrease in stigmatisation of people with Epilepsy. |
| Brabcová et al. (2013) [40] | To compare the effectiveness of educational animated video and educational drama in improving the knowledge of epilepsy and reducing epilepsy-related stigma among children aged 9–11 years | Czechia | 1,162 (Children aged 9–11, 48% female) | 7 Item Epilepsy Questionnaire | Film & Theatre by in person event through both passive and active participation in a one off campaign | Statistical | For both the educational video and educational drama there was a significant improvement in knowledge of Epilepsy, with participants doing worse (but better than baseline) at 6 month follow up. The film did significantly better in some aspects of knowledge compared to the drama. For both the educational video and educational drama there was a significant improvement in stigma towards Epilepsy, with participants doing worse (but better than baseline) at 6 month follow up. The film did significantly better in some aspects of stigma compared to the drama. |

*(Continued)*

| Author | Aim | Country | Participants/sample size | Data collection method | Art used/ Medium of delivery/level of engagement/ campaign frequency | Analysis techniques | Key findings |
|---|---|---|---|---|---|---|---|
| Dupuis et al. (2016) [41] | To explore the use of a community-based, critical arts-based project to interrogate the tragedy discourse and construct an alternative narrative of dementia using the arts | Canada | 37 Sample Size (8 People with Dementia, 7 family members, 7 visual and 8 performance artists, and 7 researchers, 70% female) | Interviews | Painting, Dance by in person event in an active participation in a one off campaign | Thematic | People with dementia were able to connect with others based on a mutual understanding of their experiences, they felt empowered by their art and wanted to share these experiences and represent dementia in a new light. An artist highlighted how much the session taught them about dementia and reported feeling more comfortable being around people with dementia, another said they were surprised by how joyous the experience was. People with dementia report feeling less internalised stigma and feeling more comfortable and stronger about their dementia. The artists involved felt the session was valuable and "walked away with personal change". Made the people with dementia feel like they can do things, creating art reduced those feelings of being trapped "you still have to thrive and look forward" |
| Garrie et al. (2016) [42] | To explore the impact of nonclinical interventions on medical students' perceptions about people with Alzheimer's disease and related dementias | USA | 11 Sample Size (Medical Students, 27% female) | Dementia Attitudes Scale (DAS), Interviews | Poetry by in person event through active participation in a one off campaign | Statistical, Thematic | Significant improvement on the DAS following the poetry session, questions such as being familiar with dementia, people with dementia being able to feel emotions, and understanding that we can improve the lives of people with dementia. Students felt more comfortable in approaching and talking to people with dementia, this is reflected in DAS question "feel confident around people with ADRD" and "enjoy interacting with people with ADRD". Students reported challenging their preconceived notions about dementia. Following the poetry session people felt more positive and many reported feeling hopeful. Felt amazed by the novel and heartful contributions of people with dementia to the poetry. |

*(Continued)*

**Table 1.** (Continued)

| Author | Aim | Country | Participants/sample size | Data collection method | Art used/ Medium of delivery/level of engagement/ campaign frequency | Analysis techniques | Key findings |
|---|---|---|---|---|---|---|---|
| George et al. (2012) [43] | To understand how the attitudes of medical students towards people with Alzheimer's disease and related dementias can be affected by participation in TimeSlips | USA | 22 Sample Size (Fourth year medical students, average age of 26.5, 45% female) | Dementia Attitudes Scale (DAS), reflective writing, focus group | Story telling by in person event through active participant participation with four sessions | Statistical, Thematic | Significant improvement on the DAS following the creative story telling session, questions such as being familiar with dementia, people with dementia can enjoy life, and understanding that we can improve the lives of people with dementia. Overall, a significant improvement in knowledge post intervention. Overall students felt significantly more comfortable in approaching and talking to people with dementia, this is reflected in DAS question "feel confident around people with ADRD" and "felt rewarded by working with people with ADRD". Students reported that their worries prior to the session were not realised and that instead they had fun. Students found the sessions to be fun and that they left feeling excited and looking forward to the next session (there were four sessions). They were surprised by the creativity of people with ADRD and felt the stories they told contributed to the excitement. |
| Gubner et al. (2020) [44] | To understand how and why the arts affect attitudes about and engagement with people with dementia through examining a service-learning general education undergraduate course centring on music, filmmaking, and dementia. | USA | 52 Sample Size (Undergraduate students, 54% were majoring in health sciences while others were doing non-health science related majors) | Reflective essays, Course evaluations | Music and Film in person event through active participation with Storytelling 6/7 sessions lasting 3 hours each | Thematic | This study identifies reproducible ways in which undergraduate arts courses thematically focused on dementia not only transform student perceptions about dementia but change the ways in which those students choose to engage with PWD (People with dementia) following course completion. Music helps students connect with people living with dementia in meaningful ways (A student shared how music visibly reduced anxiety in a participant: *"It was amazing to see how Mitch reacted to the music. I was amazed to finally see the effect that this program had on those with dementia."*), filmmaking offers students the opportunity to share unique, person-centered stories about dementia (*"We realized that Glen has such an incredible story to tell, and that in order to make a meaningful film, he didn't need to have this big reaction."*) and music that empower the voices of PWD |

*(Continued)*

| Author | Aim | Country | Participants/sample size | Data collection method | Art used/ Medium of delivery/level of engagement/ campaign frequency | Analysis techniques | Key findings |
|---|---|---|---|---|---|---|---|
| Lokon et al. (2017) [45] | This study explored the effects of participating in an intergenerational service-learning program called Opening Mind through Arts (OMA) on college students' attitudes toward people with dementia. The program is aimed at promoting the social engagement, autonomy, and dignity of people with dementia through the experience of creative self-expression. | USA | 156 Sample Size (College Students aged 18–45 with an average age of 21, 85% female) | Dementia Attitudes Scale (DAS) | Painting, Visual Arts by in person event and exhibition through active participation with 1 hour each week for 10–12 weeks | Statistical | The study results revealed a significant improvement in students' overall attitudes, comfort level, and attitude toward people with dementia. The results highlight the importance of intergenerational service learning in improving college students' overall attitudes and increase their confidence and comfort working with people with dementia. OMA service equally improved learning experience that assisted reduced students' fear and frustration and increased their confidence and comfort in interacting with people with dementia. Students also learned that people with dementia are diverse, can be creative, can enjoy life, and can feel others' kindness. |
| Roberts & Aida Farhana (2010) [46] | The study aim was focused on effectiveness of a knowledge-based intervention in increasing levels of knowledge about epilepsy and reducing negative stereotypes toward people with epilepsy through the use of first aid educational video | Australia | 131 Sample Size (Psychology Undergraduate Students aged 17–41 with an average age of 20, 56% female) | Familiarity, Attitudes, and Knowledge of Epilepsy Questionnaires | Film by video through passive participation in a one off campaign | Statistical | The study results indicated that the video (which incorporates affective testimonials of people with epilepsy as well as their family and friends, it involved engaged respondents emotionally with the issue of stigma.) was effective in both enhancing the level of knowledge of epilepsy and improving attitudes toward epilepsy. The video was rated highly on measures of acceptability. This study illustrates the effectiveness of using a simple, cost-effective educational intervention, with specific knowledge-based information incorporating the viewer's emotional involvement, to improve knowledge and reduce stigma regarding epilepsy in a targeted influential group. However, first aid video was not effective in reducing stigma concerning outcomes associated with epilepsy |

*(Continued)*

**Table 1.** (Continued)

| Author | Aim | Country | Participants/sample size | Data collection method | Art used/ Medium of delivery/level of engagement/ campaign frequency | Analysis techniques | Key findings |
|--------|-----|---------|--------------------------|------------------------|---------------------------------------------------------------------|---------------------|--------------|
| Sajatovic et al. (2017) [47] | The aim of the study was to test two new communication approaches targeting epilepsy stigma versus an education-alone approach. The study sought to evaluate whether targeted messaging strategies with one emphasizing role competence and the other emphasizing social inclusion, could effectively reduce stigma compared to a control video that simply provided knowledge about epilepsy. | USA | 295 Sample Size (Young People aged 18–29) | The Attitudes and Beliefs about Living with Epilepsy scale (ABLE), The Epilepsy Knowledge Questionnaire (EKQ) | Film by video through passive participation in a one off campaign | Statistical | The study noted that increasing knowledge alone was not enough to eliminate stigma, as negative attitudes often stem from deeper emotional and social biases rather than simple misinformation. Overall, respondents felt videos impacted their epilepsy attitudes. EKQ scores were similar across videos, with a trend for higher knowledge in experimental videos versus control (p = 0.06). The role competency and control videos were associated with slightly better perceived impact on attitudes. There were no differences between videos on ABLE scores (p = 0.568). There were subgroup differences suggesting that men, younger individuals, whites, and those with personal epilepsy experience had more stigmatizing attitudes. Additionally, video format and length play a crucial role in audience engagement. The social inclusion video, which was the longest, had the highest dropout rate, whereas the shorter role competency video maintained better engagement and was rated as more impactful. |
| Zheng et al. (2016) [48] | The study aimed at examining the extent of the stigma and explored the impact of media through a culturally tailored short film to modify dementia stigma. | USA | 90 Sample Size (Chinese Americans, 63% female) | Stigma Questionnaire | Film by in person event through passive participation in a one off campaign | Statistical, Thematic | The study results indicated that 89% (n = 80) of the randomly selected participants acknowledged that the short film was a useful way to modify their misconceptions about dementia. The use of a familiar story setting allows the audience to empathize with the message delivered via the short film. This signifies that a culturally tailored short film demonstrated promising impact in modifying stigma toward dementia. |

*(Continued)*

**Table 1.** (Continued)

| Author | Aim | Country | Participants/sample size | Data collection method | Art used/ Medium of delivery/level of engagement/ campaign frequency | Analysis techniques | Key findings |
|---|---|---|---|---|---|---|---|
| Iskander et al. (2024) [49] | The study aims to test the impact of a 3-min online video on the knowledge of stroke and factors influencing the knowledge score in four Arab countries. | Egypt, Jordan, Lebanon, UAE | 2,721 Sample Size (General population aged 25–55+, 55% female) | Knowledge of Stroke Questionnaire | Film by videos through passive participation in a one off campaign | Statistical | A significant improvement was noted in the total knowledge score in all countries from a mean average ($M_{pretest}$ = 21.11; $M_{posttest}$ = 23.70) with $p < 0.001$. Identification of the stroke risks ($M_{pretest}$ = 7.40; $M_{posttest}$ = 8.75) and warning signs ($M_{pretest}$ = 4.19; $M_{posttest}$ = 4.94), understanding the preventive measures ($M_{pretest}$ = 5.27; $M_{posttest}$ = 5.39) and the importance of acting fast ($M_{pretest}$ = 0.82; $M_{posttest}$ = 0.85) improved from baseline with ($p < 0.001$) for all score components. The educational tool successfully enhanced public understanding of stroke risks, the identification of stroke signs, and the critical need for emergency action. |
| Meira et al. (2018) [50] | The study was intended to assess the stroke knowledge of an urban population in Belo Ho-rizonte, Brazil through watching a video that consisted of a person presenting stroke signals and then they were asked to answer questions about the condition shown on the video. | Brazil | 703 Sample Size (Urban population, 62% female) | Questionnaire covering knowledge of Stroke | Film by in person event through passive participation in one off campaign | Statistical | Following the video, participants were more knowledgeable around how to support someone following a stroke. |
| Burns et al. (2018) [51] | The study aimed at examining the influence of self-revelatory theatre performance on the audience perception on Alzhei-mer's disease and dementia to determine their emotional affects, stigma reduction and increased appreciation on the utility of this type of theatre in large-scale public health education efforts. | USA | 128 Sample Size (General population with an average age of 57, 72% female) | Question covering comfort with dementia and emotions towards dementia | Theatre by exhibition through passive performance in a one off campaign | Statistical | A significant change in emotional affect from an initial strong negative affect to slightly more positive/relaxed view was noted after viewing the performance. The mean of the "Importance of Creative Arts" item was 4.53/5 at pre-performance and 4.77/5 at post-performance assessment. Findings support self-revelatory theatre as a resource to destigmatize preconceived notions of dementia. Large-scale community health education efforts could benefit from using this style of theatre to elicit a change in audience perception of disease realities. We found that overall; viewers were emotionally influenced by a single performance. They also expressed improved comfort with the idea of discussing dementia. The transformative nature of art performances is supported through the significant shift of audience perception after viewing "Seven Stages Seven Stories". |

*(Continued)*

**Table 1.** (Continued)

| Author | Aim | Country | Participants/sample size | Data collection method | Art used/ Medium of delivery/level of engagement/ campaign frequency | Analysis techniques | Key findings |
|---|---|---|---|---|---|---|---|
| Kontos et al. (2018) [52] | To illustrates the effectiveness of research based drama: *Cracked* in reducing stigma by: decreasing health care practitioners' and family carers' prejudice, fostering critical reflection about relational practices, and fostering a commitment to individual and collective action to address stigma. | Canada | 602 Sample Size (General population, Healthcare Professionals, People with Dementia, Caregivers) | Open-ended Survey Questions | Theatre by exhibition through passive performance in a one off campaign | Thematic | The study findings add to the growing body of evidence that demonstrates that research-based drama is an effective pedagogical strategy to challenge stigma associated with dementia. More specifically, study findings suggested that Cracked was effective in shifting perceptions about persons living with dementia from dominant stereotypes to a more life-affirming perspective. Further, Cracked was effective in raising awareness of the importance of bringing relationships to the forefront in conceptions of dementia and approaches to caring, and in fostering expressed commitments to challenge stigma and embrace the principles of relational caring. |
| Burns et al. (2021) [53] | To develop, implement and evaluate the impact of a short education intervention using visual arts on the understanding of dementia through visual art by primary school-aged children which aimed to increase children's understanding of dementia, reduce stigma, and encourage dementia-friendly communities by integrating dementia education into the Personal Development, Health and Physical Education (PDHPE) and Creative Arts curriculum | Australia | 74 Sample Size (Children Aged 7–10 Years Old) | Kids Insight into Dementia Survey (KIDS), Content analysis of the art created | Visual Arts by in person event and exhibition through active participation with three lessons | Statistical, Thematic | The study demonstrated that art-based education effectively improves dementia knowledge and reduces stigma among primary school children. Through visual arts, students expressed their understanding of dementia, particularly memory loss and cognitive decline. Matched pre and postintervention survey data showed a significant improvement in seven domains. This indicated that the students had an increased understanding of dementia and its impact on the individual following the intervention. It is likely that by educating children about dementia, we have the potential to reduce the stigma faced by people living with dementia and their carers. |

*(Continued)*

**Table 1.** (Continued)

| Author | Aim | Country | Participants/sample size | Data collection method | Art used/ Medium of delivery/level of engagement/ campaign frequency | Analysis techniques | Key findings |
|---|---|---|---|---|---|---|---|
| Cicero et al. (2020) [54] | To evaluate an educational comic book-based strategy to improve knowledge, attitudes, and practices about epilepsy among schoolchildren in rural and urban areas of the Bolivian Gran Chaco | Bolivia | 83 Sample Size (Children with an average age of 16, 54% female) | Knowledge, Attitudes and Practice (KAP) Questionnaire | Comics by in person event through passive participation in a one off campaign | Statistical | Significant improvements on knowledge around the cause and recognition of epilepsy, yet performance decreased as more participants understood epilepsy to be communicable through touch or breathing the same air. Responses to witnessing a seizure improved although a quarter of people would take a person with epilepsy to a traditional healer instead of a physician. Minimal participants reported negative behaviours following the intervention such as reduced avoidance behaviours and improved attitudes around people with epilepsy marrying and participating in activities. |
| Mioramalala et al. (2021) [55] | To assess the effect of a single reading of the comic book "Ao Tsara" on the knowledge, attitudes, and practices towards epilepsy of children in Malagasy schools. | Madagascar | 244 Sample Size (Children Aged 8–12+, 42% female) | Knowledge, Attitudes and Practice (KAP) Questionnaire | Comics by in person event through passive participation in a one off campaign | Statistical | Significant improvements in knowledge with less participants thinking epilepsy was contagious. Improvements in practice around being friends with people with epilepsy, supporting people with epilepsy, and not excluding people with epilepsy from class. |

or seizure. All studies using videos aimed to assess the effectiveness of educational materials. Of the 54% of studies leveraging lived experience to tackle stigma, a more diverse range of arts were used, such as visual media (54%) [34,38,44,45,47,48,53] (e.g., short films, adverts, and use of images such as comic books or posters), dance and theatre (38%) [34,36,41,51,52], auditory means (23%) [42–44] such as music, storytelling, and poetry, and painting (15%) [41,45]. Regarding the evaluation of interventions, 63% used quantitative methods [32–35,37,39,40,45–47,49–51,54,55], 21% used qualitative methods [36,38,41,44,52,56], and 17% used mixed methods [42,43,48,53].

## Impact of art interventions in reducing stigma and mechanisms of change

Only 21% [34,43–45,53] of studies involved multiple sessions of an intervention with most studies being a one off intervention which limits the conclusions which can be drawn from the effect of intervention length. 96% of included studies found positive impacts on stigma following an art intervention. These studies included a pre- and post- comparison or comparison to 'naïve' controls. The one study without a significant positive impact may be due to the control group receiving a non-art based intervention [47]. This indicates that future studies should explore the comparison between art and non-art-based interventions to fully investigate how art itself may mediate outcomes. Additionally, another study resulted in overall positive effects, however, some outcomes worsened which indicates the need to quality control the knowledge being shared during art interventions [54].

Included studies showcased diverse mechanisms of change, with 71% studies focusing on the change in attitudes and perceptions [32–34,37,39,40,42–48,51,52,54,55], 54% on raising awareness and enhancing knowledge [32–35,37–40,46,47,49,50,54,55], and 42% on the influence of emotional engagement, empathy, and social inclusion [33,36,38,41–46,51]. Studies also captured how the intervention influenced behaviours and practices (21%) [34,44,52,54,55], the use of a follow-up to demonstrate the long-term effects of art-based interventions (13%) [35,40,54], and the value of cultural adaptation (5%) [48]. This highlights the broad and dynamic ways in which arts-based interventions can tackle stigma. In the subsequent sections of the results, we use the three domains of stigma (knowledge, attitudes, behaviours) to expound on mechanisms which brought about stigma reduction. Within the theme of behaviour, we also describe the impact of the intervention on emotions,

**Shifting attitudes.** A shift in in attitudes was primarily tackled in two ways, either by educational means in which participants learnt about a disorder (79%) or by working directly alongside people with lived experience while creating art, which allows for biases and misconceptions to be rewritten (21%). This was primarily achieved through emotional engagement and fostering empathy. Theatre-based interventions helped transform negative biases into more positive perceptions, with participants reporting more relaxed and excited emotions following the performance [51]. Similarly, a 10-minute first aid informational video on epilepsy shown to psychology undergraduate students in Australia addressed stigma with significant improvements to a questionnaire around being afraid (r = .53) or nervous (r = .38) around a person with epilepsy, and led to students suggesting they would not avoid someone with frequent seizures (r = .43) [46]. Despite no direct contact with people with lived experience, there was a sense of increased understanding and appreciation of lived experience. Following an educational video and drama aimed at improving knowledge and reducing stigma towards epilepsy in children aged 9–11 years old in Czechia, the intervention group were more likely to say they would be friends with a person with epilepsy compared to the control group (81–84% vs 59%) [40]. Additionally, one study explored the utility of a live performance to tackle stigma against people with dementia and was performed at several care homes, conferences, and public theatres in Canada [52]. After the performance, one care home staff said *"we're not just caring for bodies but for people, institutions can become very mechanical"* (p. 97) and another indicated dedication to behavioural change following the performance saying *"I'm going to try living these ideas"* (p. 97), which demonstrates the power of the arts to reduce stigma [52].

Of the 21% of studies that involved the co-production of art alongside a person with lived experience, the experience of participants was positive [41–45]. Interestingly, all co-production occurred alongside people with dementia. Co-production activities ranged from 1–12 sessions depending on the study and 80% of co-production studies focused the intervention towards university students. In a creative story telling intervention with medical students and people with dementia in the USA, after four sessions, there were significant improvements in feeling relaxed around people with dementia, no longer feeling afraid of people with dementia, and being familiar with dementia [43]. The students discussed how *"None of the worries I had previous to our first day manifested themselves, it was a very fun experience."* (p. 325) and *"I think seeing the patients here humanised them… it's not just slapping a label on a person."* (p. 326) [43]. These findings were similar among another study which involved a single art workshop with medical students and people with dementia where they co-created poetry and compared descriptions of dementia pre- and post- workshop, being *"Quiet, reserved, frightened, moody"* to *"Kind lively, happy, and funny"* respectively (p. 5) [42]. One study focussed on the artist's experience following the co-production of paintings and a dance routine in a single session alongside people with dementia [41]. The artists reported feeling *"more comfortable being around people with dementia"* (p. 366) and *"felt surprised that it was such a positive experience"* (p. 366) [41]. Although it can be difficult to untangle whether art or the act of working alongside a person with lived experience drove these findings.

A handful of studies explored the experiences of persons with lived experience in participating in the art intervention, whether they were the audience or were involved in the co-production. In an arial silk performance in Brazil, people with epilepsy felt valued and represented when viewing the performance and related to several scenes [36]. In the

**Table 2. Patterning chart of included studies.**

| Author | Country | | | | | | Disorder | | | Intervention Target | | |
|---|---|---|---|---|---|---|---|---|---|---|---|---|
| | North America | South America | Europe | Asia | Australasia | Africa | Epilepsy | Dementia | Stroke | Children (under 18 years) | University Students | General Population |
| Fong et al. (2018) [32] | | | | X | | | X | | | X | | X |
| Fong et al. (2019) [33] | | | | X | | | X | | | | | X |
| Haesebaert et al. (2020) [34] | | | X | | | | | | X | | | X |
| Mikulik et al. (2019) [35] | | | X | | | | | | X | X | | |
| Spagnol et al. (2019) [36] | | X | | | | | X | | | | | X |
| Tekle-Haimanot et al. (2016) [37] | | | | | | X | X | | | X | | |
| Zhao & Liu (2021) [38] | X | | X | X | | | | | X | | | X |
| Brabcová et al. (2017) [39] | | | X | | | | X | | | X | | |
| Brabcová et al. (2013) [40] | | | X | | | | X | | | X | | |
| Dupuis et al. (2016) [41] | X | | | | | | | X | | | | X |
| Garrie et al. (2016) [42] | X | | | | | | | X | | | X | |
| George et al. (2012) [43] | X | | | | | | | X | | | X | |
| Gubner et al. (2020) [44] | X | | | | | | | X | | | X | |
| Lokon et al. (2017) [45] | X | | | | | | | X | | | X | |
| Roberts et al. (2010) [46] | | | | | X | | X | | | | X | |
| Sajatovic et al. (2017) [47] | X | | | | | | X | | | | X | X |
| Zheng et al. (2016) [48] | X | | | | | | | X | | | | X |
| Iskander et al. (2024) [49] | | | | X | | X | | | X | | | X |
| Meira et al. (2018) [50] | | X | | | | | | | X | | | X |
| Burns et al. (2018) [51] | X | | | | | | | X | | | | X |
| Kontos et al. (2020) [52] | X | | | | | | | X | | | | X |
| Burns et al. (2021) [53] | | | | | X | | | X | | X | | |
| Cicero et al. (2020) [54] | | x | | | | | x | | | x | | |
| Mioramalala et al. (2021) [55] | | | | | | x | x | | | x | | |

co-production of paintings and a dance routine in Canada, people with dementia were able to see dementia in a different light which brought a sense of comfort and reduced internalised stigma [41].

**Raising awareness and enhancing knowledge.** Participation in art interventions allows participants to become more familiar with the disorder, whether through watching videos or through creating art alongside with people with lived experience. Enhancing knowledge stood out as a prominent mechanism, particularly using visual and narrative art forms. For example, one study used educational videos and quizzes with students aged 13–19 years old and teachers in Malaysia which led to substantial improvements in knowledge of epilepsy, with knowledge scores increasing from low and moderate respectively to very high [32]. Another study which used comic books to tackle stigma of epilepsy in Ethiopia not only boosted understanding of causes and whether the disorder was contagious, but also improved practical skills, with accurate first-aid responses for seizures rising from 30% to 68% in the questionnaire [37].

The arts were not just effective at improving knowledge but also facilitating an understanding of lived experience. For example, an arts program involving the co-development of visual art projects with undergraduate students and people with dementia in the USA reported positive effects in understanding that people with dementia can enjoy life (r = −.4) and can

| People with lived experience | Population Wide | Educational Materials | Sharing Lived Experience | Videos | Painting | Dance and Theatre | Visual Media (Short Films, Photography) | Auditory (Music, Story Telling, Poetry) | Quantitative | Qualitative |
|---|---|---|---|---|---|---|---|---|---|---|
| | | X | | X | | | | | X | |
| | | X | | X | | | | | X | |
| | x | X | X | X | | X | X | | X | |
| | | X | | X | | | | | X | |
| X | | | X | | | X | | | | X |
| | | X | | | | | X | | X | |
| | x | X | X | | | | X | | | X |
| | | X | | X | | | | X | X | |
| | | X | | X | | X | | | X | |
| X | | | X | | X | X | | | | X |
| | | | X | | | | | X | X | X |
| | | | X | | | | | X | X | X |
| | | | X | | | | X | X | | X |
| | | | X | | X | | X | | X | |
| | | X | | X | | | | | X | |
| | | X | X | | | | X | | X | |
| | | X | X | | | | X | | X | X |
| | X | X | | X | | | | | X | |
| | X | X | | X | | | | | X | |
| | | | X | | | X | | | X | |
| X | | | X | | | X | | | | X |
| | | | X | | | | X | | X | X |
| | | x | | | | | x | | x | |
| | | x | | | | | x | | x | |

feel when others are kind to them (r = −.37) [45]. In a movie review of a short film exploring an older couple's lived experience of dementia tailored for Chinese Americans, one comment read *"now I know dementia is not about going crazy"* [48]. In another study, Australian children aged 8–10 years old attended three lessons, one to create a painting of a memory, then to learn about dementia, and lastly to recreate the original art through the lens of someone with dementia [53]. The latter artworks were noticeably darker in mood and more abstract in shape which demonstrated an increased awareness and understanding of dementia. These studies exemplify that the arts can be used as a medium to raise awareness of disorders, not just of their existence but also to reframe representations.

**Behavioural and emotional impact.** Behavioural and emotional changes were achieved through interventions that developed practical skills and encouraged cognitive reframing. In some studies, the value of art as a tool to tackle stigma lay in its ability to elicit emotional reactions. A range of studies report participants feeling emotional reactions to the art, such as the review of a short film aimed at raising stroke awareness: *"the film is very profound, moving, and beautiful."* (Supplementary S3 File) [38]. Storytelling and creative expression initiatives which involved four sessions enabled medical students to adopt a more person-centred approach, with many noting an improved ability to see beyond the

diagnosis in clinical environments "*Every person is different. So just by saying you have AD that shouldn't mean you are the same as other people*" (p. 326) [43]. Direct engagement with people with lived experience was found to be fun, engaging, exciting, and heartfelt [42,43]. In one study, 6 or 7 art sessions alongside people with dementia led to 56% of participants continuing engagement in some way beyond the initial course either through continuing volunteering post course completion, shifting toward a geriatric focus in clinical work, or integrating the skills learnt into the care of people with dementia in their own families [44]. Quantitative data from another study which involved a county wide awareness campaign in France spanning two months led to a 15% increase in emergency calls for stroke symptoms, demonstrating enhanced public responsiveness [34]. This demonstrated the behavioural and emotional influences of art in addressing stigma, particularly in reference to working directly alongside people with lived experience and in watching short films and performances.

### Benefits and challenges to art-based interventions

Some of the reviewed articles indicated facilitators and barriers to the sustainability of arts-based interventions for neurological disorders. A key facilitator was the adaptability of art forms, allowing interventions to accommodate diverse participant needs. For instance, offering varied artistic mediums such as educational videos and storytelling enabled individuals with different learning preferences to engage effectively, promoting sustained interest and reducing stigma [39]. Emotional and social support also played a crucial role, with collaborative art projects helping participants build social connections and reduce feelings of isolation. Involving people with dementia in shared artistic experiences fostered a sense of belonging and mutual understanding [41]. Additionally, interventions that resonated personally with participants enhanced motivation and long-term engagement, where participants expressed feeling valued when their emotions were artistically represented [36]. Only one article explored culturally tailored interventions and found this effectively reduced stigma in specific cultural contexts, with 89% of Chinese American participants reporting shifts in their perceptions of dementia [48]. However, minimal details were reported on what constituted tailoring this intervention for a different culture.

Utilising the arts can provide a vehicle to spread awareness of a disorder. In a study using a short film to promote stroke awareness, one of the movie reviews noted *"The very first thing I did after seeing the movie was forward it to my father!"* (Supplementary S3 File) [38]. However, certain barriers also impacted the sustainability of these interventions. Physical and cognitive limitations among participants sometimes hindered consistent engagement, particularly in programs with high physical demands [35]. Resource availability emerged as a significant challenge, with limited funding affecting the reach and effectiveness of a stroke awareness campaign in France which used visual arts [34]. The burden on caregivers was another obstacle, where caregivers often struggled to balance their responsibilities, potentially affecting the continuity of interventions [36]. Moreover, measuring the long-term impact of arts-based interventions proved difficult. One study showed that while short films could modify stigma effectively, quantifying their lasting influence on community attitudes remained challenging [48].

## Discussions

### Key findings

The review found that arts-based interventions effectively contributed to stigma reduction in neurological disorders through three primary mechanisms: shifting attitudes, raising awareness thus enhancing knowledge, and influencing behaviours. Additionally, some studies illustrated that through evoking emotional responses, desirable changes such as improvement of discriminatory attitudes was observed. Knowledge, attitudes and behaviours are widely used constructs of stigma in mental and brain health, but emotions are a less explored construct, yet the findings suggest that they might mediate stigma reduction in some participants. Future studies should explore the role of emotions in imparting change alongside other established constructs of stigma.

Included studies explored art-interventions for reducing stigma in epilepsy, stroke, and dementia. While these are important disorders to address, it is important to consider additional neurological disorders such as Parkinson's disease and multiple sclerosis that are both highly prevalent and stigmatised [1,17,57]. These disorders can be managed well through appropriate treatment, yet, especially for those living in LMICs countries, stigma acts as a barrier to accessing healthcare [10,58]. This highlights the lack of attention within the current literature and suggests an important gap. Art-based interventions provide a unique opportunity to tackle stigma through fostering public understanding and empathy which may promote early diagnosis and management [5].

The finding on the utility of visual and narrative arts (such as the use of comic books) highlighted potential artistic approaches for tackling stigma among young populations such as school going children, university students and crucial demographics like medical students who eventually become part of healthcare systems. Early intervention has a high potential of bringing sustained positive change but the mechanism through which this change is sustained in unclear [59]. One possible explanation is that early exposure to visual and narrative arts creates an immersive experience delivered through experiential learning which fosters greater empathy than didactic approaches alone [60]. However more robust testing is required across multiple neurological disorders and multiple settings. Secondly, considerations such as cost-effectiveness and potential for scaling up should be made. Digital media, while not providing direct interactions with the public may be a potential platform for scaling up such interventions as they allow for an accessible, cost-effectiveness, and large reaching strategy which can facilitate quick comprehension and retention of critical information [61].

Regarding evaluation of effectiveness of these interventions, most studies used one-off, cross-sectional questionnaires or interviews. Behaviour change is difficult, takes time and can may not be sustained long-term. While a small number of studies followed up participants post-intervention and/or investigated changes to behaviours (primarily through self-reported behaviour questions), overall, the depth of data collected limits the understanding of the influence interventions may have on stigma, and the particular components of the intervention that contributed to reducing stigma. Self-reported responses encourage a social desirability bias leading to responses which may not accurately reflect the behaviours and attitudes of a participant [62]. To capture changes in stigma on a deeper level, methods such as a reflective diary may be more appropriate, as well as longitudinal follow-up.

There was a significant gap in the use of art-based interventions for neurological awareness in LMICs with only two countries in sub-Saharan Africa being represented(Ethiopia and Madagascar). Most of the reviewed studies were conducted in high-income Western countries, with limited representation from South America, Asia, and the Middle East. This is of issue as the need for anti-stigma interventions is higher in these regions [12,13] with LMICs containing the majority of burden from disease within the world [2]. Yet the amount of funding and human resources needed to adequately address healthcare issues brought about by stigma is consistently unmet and undervalued within LMICs. Additionally, this geographical and cultural paucity limits our understanding of contextual factors which could influence the effectiveness of art-based interventions while also highlighting a critical need for expanding art-based interventions to underrepresented regions. For example, a scoping review of art projects in sub-Saharan Africa found a primary focus on HIV and AIDS which has successfully raised awareness and empowered individuals to take control of their own health [63] and these approaches can be extended to brain health. Additionally, limited efforts have been made in tailoring interventions for different cultures with minimal details around how best to ensure the relevancy of art based interventions to different cultural and religious backgrounds.

## Strengths and limitations

We used the PRISMA-SR guidelines to undertake this review which ensured methodological rigour and minimised bias. By involving at least two reviewers at both the screening and data extraction stage, the risk of bias was reduced [64,65]. By screening 9,990 articles, this review provides a thorough overview of available literature, however, we acknowledge key information may have been missed from conference abstracts and activities to raise awareness and tackle stigma

occurring outside of academic research [66]. Hence, while this review provides a strong overview of available literature, we acknowledge the limitation of being unable to include non-academic art interventions and those without explicit evaluations, such as short films and creative art workshops within communities [67–70]. This review includes studies that possess both a participatory nature, as well as the dissemination of knowledge and experiences through the arts and media. Due to the varying elements within these two approaches, there may be differences in results, however, we have accounted for this by distinguishing between the two during the synthesis of results.

Reporting how gender, intervention length, and cultural context influence the effectiveness of art-based interventions was limited during this review due to the scarcity of literature exploring these topics. Additionally, questions around how stigma interventions interact within the wider context of an individual, such as family members, would provide an opportunity to better understand the spread of awareness raising efforts. To better understand how best to deliver and who best to deliver art interventions to, future art-based interventions tackling stigma should explore different audiences and delivery lengths to improve upon current knowledge.

## Conclusion

This review provided a methodologically rigorous overview on the role of art in tackling stigma in neurological disorders. It demonstrated that art-based interventions can be a powerful tool in reducing stigma around neurological disorders, both in terms of raising awareness and shifting attitudes towards neurological disorders. Interventions prioritising the direct interactions between a small number of the public and people with lived experience seems particularly effective at driving meaningful and lasting changes in attitudes.

More robust evidence of effectiveness of these approaches across diverse settings and a diverse range of highly stigmatised disorders such as Parkinson's disease and multiple sclerosis is required. Lastly, robust and enduring mechanisms of measuring effectiveness of these interventions using mixed methods data and models of how these interventions bring change should be explored.

## Supporting information

**S1 File. Completed PRISMA Checklist for the systematic review.**
(DOCX)

**S2 File. Search Strategy.**
(DOCX)

**S3 File. Standardized data extraction template applied to included studies.**
(XLSX)

## Author contributions

**Conceptualization:** Mary A. Bitta, Natasha Fothergill-Misbah.

**Data curation:** Mary A. Bitta, Vivian Nyadimo, Shilla Dama Unda.

**Formal analysis:** Jack Lumsdon, Vivian Nyadimo, Shilla Dama Unda.

**Funding acquisition:** Mary A. Bitta, Natasha Fothergill-Misbah.

**Investigation:** Natasha Fothergill-Misbah.

**Methodology:** Mary A. Bitta, Jack Lumsdon, Vivian Nyadimo, Natasha Fothergill-Misbah.

**Project administration:** Vivian Nyadimo, Natasha Fothergill-Misbah.

**Resources:** Mary A. Bitta, Natasha Fothergill-Misbah.

**Supervision:** Natasha Fothergill-Misbah.

**Validation:** Jack Lumsdon, Vivian Nyadimo, Natasha Fothergill-Misbah.

**Visualization:** Mary A. Bitta, Jack Lumsdon.

**Writing – original draft:** Jack Lumsdon, Vivian Nyadimo, Shilla Dama Unda.

**Writing – review & editing:** Mary A. Bitta, Jack Lumsdon, Vivian Nyadimo, Natasha Fothergill-Misbah.

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
