## [Decision Letter · Decision Letter 0]

20 Aug 2025

Dear Dr. Bitta,

plosone@plos.org . . A rebuttal letter that responds to each point raised by the academic editor and reviewer(s). You should upload this letter as a separate file labeled 'Response to Reviewers'.A marked-up copy of your manuscript that highlights changes made to the original version. You should upload this as a separate file labeled 'Revised Manuscript with Track Changes'.An unmarked version of your revised paper without tracked changes. You should upload this as a separate file labeled 'Manuscript'.

We look forward to receiving your revised manuscript.

Kind regards,

Loretta Giuliano, M.D.

Academic Editor

PLOS ONE

Journal Requirements:

https://journals.plos.org/plosone/s/file?id=ba62/PLOSOne_formatting_sample_title_authors_affiliations.pdf ..

“This project was funded by the British Academy grant number OIIRP230192”

Reviewers' comments:

Reviewer's Responses to Questions

**Comments to the Author**

1. Is the manuscript technically sound, and do the data support the conclusions?

Reviewer #1: Yes

Reviewer #2: Yes

Reviewer #3: Yes

2. Has the statistical analysis been performed appropriately and rigorously?

Reviewer #1: Yes

Reviewer #2: N/A

Reviewer #3: N/A

3. Have the authors made all data underlying the findings in their manuscript fully available?

Reviewer #1: Yes

Reviewer #2: Yes

Reviewer #3: Yes

4. Is the manuscript presented in an intelligible fashion and written in standard English?

Reviewer #1: Yes

Reviewer #2: Yes

Reviewer #3: Yes

Reviewer #1: This manuscript effectively addresses how lack of knowledge and misconceptions about neurological disorders often foster prejudice, such as the perception that “people with abnormal movements = people with mental illness.” The authors’ findings suggest that educational and awareness activities can shift this perception toward “people with abnormal movements = patients.” From the results presented, the following sequence of cognitive change can be inferred:

An initial recognition that the individual has a medical condition.

Activation of the moral value that it is wrong to hold negative views toward people who are ill.

Based on this moral value, acquisition of knowledge about the disease leads to a reduction in stigma.

In contrast, in low- and middle-income countries, several factors may contribute to the persistence of stigma: (i) lack of awareness about neurological diseases due to limited medical and educational resources, (ii) insufficient development of the moral framework that underpins non-discriminatory attitudes toward illness, and (iii) a high prevalence of untreated patients whose symptoms remain pronounced in daily life. A deeper analysis of these specific circumstances could strengthen the manuscript’s discussion of challenges unique to low- and middle-income settings.

Furthermore, the background and discussion presented by the authors hold important implications in the context of global migration and labor mobility, where cultural and religious backgrounds are increasingly diverse. For example, when migration limits access to religious institutions or community networks, the continuity of moral education and shared values may be disrupted, potentially hindering cross-cultural understanding of disease and the reduction of stigma. Through the framework provided by this study, exploring the relationship between changes in moral education, cultural background, and stigma could offer valuable—albeit delicate—insights from both medical and societal perspectives.

Therefore, the significance of this work lies not only in evaluating the effectiveness of art-based interventions but also in raising awareness of underlying issues related to moral education, cultural foundations, and the transformation of values in the context of international migration—issues that are contemporary, unavoidable, and worth understanding through a medical lens.

Reviewer #2: This is an interesting paper that summarizes the art-based interventions reported in literature to tackle stigma in neurological disorders.

It is a nice ad useful review that helps to give clarity and consistence to a delicate and under-valued topic.

I will try to suggest some revisions that might improve the paper.

Terminology:

•Page 5 line 55: insert (LMIC) next to the extended expression

•Page 6 lines 93-94: delete the sentence: One such approach is the use of art to challenge negative stereotypes and foster social change. (repetition)

•Page 7 line 105 delete “the” before performing arts

Concept

I was surprised to notice the paucity of data about Africa and South America. I found that some interesting papers have not been cited in your revision. For example, a comic-book based study in Bolivia (Cicero CE, et al. Comic book-based educational program on epilepsy for high-school students: Results from a pilot study in the Gran Chaco region, Bolivia. Epilepsy Behav. 2020 Jun; 107:107076. doi: 10.1016/j.yebeh.2020.107076. Epub 2020 Apr 18. PMID: 32315969.) and one study among students in Madagascar (Mioramalala SA, et al. Effects of an educational comic book on epilepsy-related knowledge, attitudes and practices among schoolchildren in Madagascar. Epilepsy Res. 2021 Oct;176:106737. doi: 10.1016/j.eplepsyres.2021.106737. Epub 2021 Aug 9. PMID: 34419769) Did you exclude these articles on purpose, and, in case, for which reasons, or your search strategy was not able to find these papers?.

Reviewer #3: The manuscript provides a well-structured review with a clear summary of the literature. I have provided comments and suggestions for improvement; overall, the review represents a valuable contribution and could benefit from minor clarifications and further discussion of future research directions.

**Do you want your identity to be public for this peer review?** For information about this choice, including consent withdrawal, please see our For information about this choice, including consent withdrawal, please see our Privacy Policy .

Reviewer #1: **Yes:** Keiichi AbeKeiichi Abe

Reviewer #2: No

Reviewer #3: No

While revising your submission, please upload your figure files to the Preflight Analysis and Conversion Engine (PACE) digital diagnostic tool, https://pacev2.apexcovantage.com/ . PACE helps ensure that figures meet PLOS requirements. To use PACE, you must first register as a user. Registration is free. Then, login and navigate to the UPLOAD tab, where you will find detailed instructions on how to use the tool. If you encounter any issues or have any questions when using PACE, please email PLOS at . PACE helps ensure that figures meet PLOS requirements. To use PACE, you must first register as a user. Registration is free. Then, login and navigate to the UPLOAD tab, where you will find detailed instructions on how to use the tool. If you encounter any issues or have any questions when using PACE, please email PLOS at figures@plos.org . Please note that Supporting Information files do not need this step.. Please note that Supporting Information files do not need this step.

---

## [Author Response · Author response to Decision Letter 1]

13 Jan 2026

Response to reviewers

Thank you for considering this manuscript for publication within PLOS One. The reviewer comments have been incredibly insightful for improving the quality of this manuscript. Please see below for a list of reviewer feedback and the changes made to the manuscript based upon this.

Update to the financial disclosure statement: “This project was funded by the British Academy grant number OIIRP230192. The funder had no role in the study design, analysis, decision to publish, or preparation of the manuscript.”

Kind regards

Mary A. Bitta

Journal Requirements:

Response: Updated

“This project was funded by the British Academy grant number OIIRP230192”

Response: Amended the statement and included in the above cover letter

Response:

As this manuscript is a review, no data has been generated, although the tables contain all necessary data needed to replicate the analysis of included studies

4. Please include captions for your Supporting Information files at the end of your manuscript, and update any in-text citations to match accordingly. Please see our Supporting Information guidelines for more information: http://journals.plos.org/plosone/s/supporting-information. U

Response: Updated the formatting of the manuscript

Response: Reviewer #2 suggested the inclusion of two studies within the review, these are both appropriate studies which would have been included had they appeared within the search, hence, they have been included within the review.

Response: Reviewed to ensure references are complete and correct. Several additional references have been added following changes to the manuscript.

Reviewer #1: This manuscript effectively addresses how lack of knowledge and misconceptions about neurological disorders often foster prejudice, such as the perception that “people with abnormal movements = people with mental illness.” The authors’ findings suggest that educational and awareness activities can shift this perception toward “people with abnormal movements = patients.” From the results presented, the following sequence of cognitive change can be inferred:

An initial recognition that the individual has a medical condition.

Activation of the moral value that it is wrong to hold negative views toward people who are ill.

Based on this moral value, acquisition of knowledge about the disease leads to a reduction in stigma.

In contrast, in low- and middle-income countries, several factors may contribute to the persistence of stigma: (i) lack of awareness about neurological diseases due to limited medical and educational resources, (ii) insufficient development of the moral framework that underpins non-discriminatory attitudes toward illness, and (iii) a high prevalence of untreated patients whose symptoms remain pronounced in daily life. A deeper analysis of these specific circumstances could strengthen the manuscript’s discussion of challenges unique to low- and middle-income settings.

Response: Points suggested help to further elaborate on the context in which stigma occurs, this has been incorporated into the second paragraph of the introduction and the final paragraph of the key findings section in the discussion to further highlight the unique factors which influence the experience of stigma in low- and middle-income countries.

Furthermore, the background and discussion presented by the authors hold important implications in the context of global migration and labor mobility, where cultural and religious backgrounds are increasingly diverse. For example, when migration limits access to religious institutions or community networks, the continuity of moral education and shared values may be disrupted, potentially hindering cross-cultural understanding of disease and the reduction of stigma. Through the framework provided by this study, exploring the relationship between changes in moral education, cultural background, and stigma could offer valuable—albeit delicate—insights from both medical and societal perspectives. Therefore, the significance of this work lies not only in evaluating the effectiveness of art-based interventions but also in raising awareness of underlying issues related to moral education, cultural foundations, and the transformation of values in the context of international migration—issues that are contemporary, unavoidable, and worth understanding through a medical lens.

Response: Really insightful point around the complexity of stigma and the multiple perspectives needed to be considered when designing interventions. Unfortunately, only a single article explored the culturally tailoring an intervention and did not provide any details about the process of tailoring an intervention to different cultural and religious backgrounds and so it is difficult to elaborate upon this point – this is mentioned in the benefits and challenges to art-based interventions section. Considering LMICs are most in need of anti-stigma interventions, further details on this point have been elaborated in the discussion at the end of the key findings section to give further context.

Reviewer #2: This is an interesting paper that summarizes the art-based interventions reported in literature to tackle stigma in neurological disorders.

It is a nice ad useful review that helps to give clarity and consistence to a delicate and under-valued topic.

I will try to suggest some revisions that might improve the paper.

Terminology:

•Page 5 line 55: insert (LMIC) next to the extended expression Added

•Page 6 lines 93-94: delete the sentence: One such approach is the use of art to challenge negative stereotypes and foster social change. (repetition)

Response: Deleted the final sentence of the previous paragraph

•Page 7 line 105 delete “the” before performing arts Deleted

Concept

I was surprised to notice the paucity of data about Africa and South America. I found that some interesting papers have not been cited in your revision. For example, a comic-book based study in Bolivia (Cicero CE, et al. Comic book-based educational program on epilepsy for high-school students: Results from a pilot study in the Gran Chaco region, Bolivia. Epilepsy Behav. 2020 Jun; 107:107076. doi: 10.1016/j.yebeh.2020.107076. Epub 2020 Apr 18. PMID: 32315969.) and one study among students in Madagascar (Mioramalala SA, et al. Effects of an educational comic book on epilepsy-related knowledge, attitudes and practices among schoolchildren in Madagascar. Epilepsy Res. 2021 Oct;176:106737. doi: 10.1016/j.eplepsyres.2021.106737. Epub 2021 Aug 9. PMID: 34419769) Did you exclude these articles on purpose, and, in case, for which reasons, or your search strategy was not able to find these papers?

Response: Thank you for recommending these studies, they are both extremely relevant to our review, hence they have both been added into the review and the results section updated as appropriate. Unfortunately, our search strategy was not able to find these articles and so thank you for bringing these to our attention.

Reviewer #3: The manuscript provides a well-structured review with a clear summary of the literature. I have provided comments and suggestions for improvement; overall, the review represents a valuable contribution and could benefit from minor clarifications and further discussion of future research directions.

The authors present a review investigating Exploring the use of art interventions in challenging stigmas related to neurological disorders: A scoping review. The study is well-structured and addresses an important topic. I have a few comments and suggestions that could help further improve the clarity of the manuscript.

Introduction

Line 69-70 the authors mentioned “These stigmas are reinforced by cultural beliefs, fear, and the visibility or unpredictability of symptoms”. The authors may consider highlighting whether certain cultural stigmas related to neurological disorders are shared across countries with different income levels.

Response: Thank you for your comments, additional details around shared cultural stigmas across different contexts have been added into this paragraph to better improve the generalisability of this review

2. Line 70-71 The article mentions that disorders like epilepsy or Parkinson’s are sometimes perceived as spiritual afflictions or witchcraft, fueling rejection. Are beliefs that disorders such as epilepsy or Parkinson’s are spiritual afflictions or witchcraft primarily reported in African countries, or do similar perceptions occur in other regions as well?

Response: Clarifications have been added that beliefs around witchcraft and curses are specific to Africa and added in that other continents such as Asia struggle with other forms of cultural specific stigmas such as secretiveness around diagnoses.

3. The authors may also consider adding brief context on how cultural stigmas influence both access to healthcare and the social inclusion of individuals with neurological disorders in different settings.

Response: Rounded out the paragraph to tie in different cultural stigmas around neurological disorders and how that impacts healthcare and wellbeing.

Methods

4. Section 2.2 - The authors mention that studies were reviewed up to October 2024. It would be helpful to indicate the starting date to clarify the full-time span of the literature considered.

Response: Clarified that articles were searched from inception to October 2024

5. Section 2.5 – The authors mention that a data extraction form was developed to collect relevant information. Could you consider making this form publicly available, as it could help other research groups conduct similar reviews?

Response: Thank you for your suggestion, as the data extraction form was simplified and streamlined to form Table 1. It seems redundant to also include the data extraction form as it includes many of the same details

Results

6. In Section 3.5, the authors describe the number of art sessions and outcomes for participants (reference 36). It would be helpful to clarify whether the reviewed studies reported the duration of each intervention, and if so, to indicate this information, as it could provide important context for interpreting the results.

Response: Data on intervention duration was extracted and is presented in Table 1 but agreed that this should be made clearer. I have written a sentence about intervention duration at the beginning of the section on impact of art interventions in reducing stigma and mechanisms of change and have included details around intervention duration where relevant throughout the results section.

Discussion

7. The authors report in the abstract that most studies were conducted in highincome settings. It would be valuable to discuss whether this pattern reflects a higher prevalence of neurodegenerative diseases in these regions or is predominantly driven by the greater availability of research resources in highincome countries.

Response: Prevalence of neurological disorders remains the same if not higher in lower- and middle-income countries, although this is commonly under reported due to poor methodological quality of prevalence studies. Most likely, fewer studies were conducted in lower- middle-income countries due to poorer availability of research resources and reduced priority of stigma interventions. This has been added into the final paragraph in they key findings section of the discussion to give further context.

8. Given that most of the cited studies did not examine gender, it could be valuable to consider hypotheses regarding how gender-related factors might influence interest or engagement in art.

Response: Gender dynamics were poorly reported on in included studies which hindered the ability to discuss this within the review, future directions have now been included around examining how gender related factors may influence the efficacy of art interventions on stigma

9. The authors could consider discussing the hypothesis that inclusive family environments support continued engagement in art-based interventions, potentially enhancing long-term quality of life.

Response: Family support structures was not a strong theme within the included articles so it would be difficult to adequately address this point with the available data although this is an interesting point that should be explored in future research

---

## [Editor Report · Decision Letter 1]

8 Feb 2026

Exploring the use of art interventions in challenging stigmas related to neurological disorders: A scoping review

PONE-D-25-31810R1

Dear Dr. Bitta,

We’re pleased to inform you that your manuscript has been judged scientifically suitable for publication and will be formally accepted for publication once it meets all outstanding technical requirements.

Kind regards,

Loretta Giuliano, M.D.

Academic Editor

PLOS One
---

## [Editor Report · Acceptance letter]

PONE-D-25-31810R1

PLOS One

Dear Dr. Bitta,

I'm pleased to inform you that your manuscript has been deemed suitable for publication in PLOS One. Congratulations! Your manuscript is now being handed over to our production team.

Kind regards,

on behalf of

Dr. Loretta Giuliano

Academic Editor

PLOS One